# Lived Experiences of Everyday Memory in Adults with Dyslexia: A Thematic Analysis

**DOI:** 10.3390/bs13100840

**Published:** 2023-10-14

**Authors:** James Hugo Smith-Spark, Elisa G. Lewis

**Affiliations:** Division of Psychology, School of Applied Sciences, London South Bank University, 103 Borough Road, London SE1 0AA, UK; lewise12@lsbu.ac.uk

**Keywords:** developmental dyslexia, adult cognition, memory, short-term memory, working memory, long-term memory, prospective memory, assistive technology, lived experience

## Abstract

Dyslexia-related difficulties with memory are well documented under laboratory conditions and via self-report questionnaires. However, the voice of the individual with dyslexia regarding the lived experience of memory across different memory systems and different daily settings is currently lacking. To address this gap in the literature, semi-structured in-depth interviews were conducted with 12 adult female university students with dyslexia. Questions probed different memory systems and experiences across different settings, with interviewees also being asked about their use of technology to support their memory. Two overarching themes were identified in the subsequent thematic analysis. The theme of fallibility of memory had two sub-themes of (i) a lack of trust and confidence in memory and (ii) factors contributing to memory failure. The second theme, facilitators of memory, also consisted of two sub-themes, relating to (i) a preference for traditional tools to support memory and (ii) the use of digital tools to support memory: benefits and limitations. The current study gives insights into the rich and complex extended and distributed cognitive systems of adults with dyslexia. The implications of the findings for dyslexia theory, support in educational and work settings as well as assistive technology development are considered.

## 1. Introduction

Developmental dyslexia (henceforth, dyslexia) is characterized by core problems with phonological processing that affect reading, writing, and spelling (for reviews, see [1,2,3]) and that vary in both the way they are manifested and their relative severity. In addition to these widely acknowledged core difficulties, dyslexia-related problems with a range of memory systems have been identified under laboratory conditions in both children (e.g., [4,5,6,7,8], although see [9] for a recent dissenting view) and adults (e.g., [10,11]). Further to this evidence, a small number of studies have used naturalistic task demands or required responses embedded in everyday or virtual reality settings (e.g., [12,13,14,15]). As well as the experimental work conducted inside and outside the laboratory, there are also self-report questionnaire data that indicate how cognitive problems (including memory difficulties) are likely to play out in the daily lives of children and adults with dyslexia [15,16,17,18,19,20,21]. Given the diverse uses to which memory is put in everyday life, dyslexia-related deficits in memory are likely to have a deleterious impact on life chances and life satisfaction. Despite these negative consequences, the lived experience of people with dyslexia in different memory systems in day-to-day settings is currently lacking. There is considerable value to be gained from documenting the lived experiences of adults with dyslexia, given the potential negative implications of memory problems across educational, work, social, and personal settings. In so doing, it should be possible to identify the areas in which there will be a greater likelihood of error by adults with dyslexia and for these areas to be officially recognized in support arrangements and reasonable adjustments to educational and working practices. The study reported in this paper was, therefore, conducted to give a greater voice to adults with dyslexia. In-depth semi-structured interviews were carried out in which the participants’ experiences with a range of memory systems were explored across different settings. The ways in which technology was enlisted by adults with dyslexia to support their memory abilities were also probed to better understand their interactions with these tools.

To ensure that such support provision is appropriately targeted in adulthood, it is important to identify how the kinds of memory problems found under controlled laboratory conditions are most likely to play out in the daily lives of individuals with dyslexia. Two ways to gain this understanding are (i) through experimentation using more naturalistic task demands or conditions (e.g., [15,21]) or (ii) through asking people with dyslexia about their experiences (either via self-report questionnaires or by interviewing them). The difference between two levels of cognition has been highlighted in the literature (e.g., [22,23], namely the algorithmic (measuring optimal performance, usually under controlled laboratory conditions) and the reflective (tending to rely on self-reports to assess typical performance in daily life). This theoretical approach highlights the importance of understanding typical, as well as optimal, performance in order to understand cognition more fully. To gain further insights into the way in which dyslexia affects cognition at the reflective level of this conceptual framework, the current paper focused on lived experiences across three different memory systems (short-term/working memory, long-term memory, and prospective memory). The documented effects of dyslexia on each of these memory systems in adulthood will now be considered briefly in turn.

Short-term memory is responsible for storing information for short periods of time. Verbal and visuospatial information stored in this system is likely to be lost after around 30 s if it is not maintained actively (e.g., [24,25]). Verbal short-term memory deficits are well documented under laboratory conditions in adults with dyslexia (e.g., [26,27,28,29,30], see [31] for a review of short-term memory in children and adults with dyslexia). While less research has been carried out on visuospatial short-term memory in adults with dyslexia, deficits do not seem to be present in this modality (e.g., [11]). In interviews about day-to-day experiences, university students with dyslexia have identified verbal short-term memory as being problematic [32]. The students highlighted remembering items on lists, telephone numbers, formulae, and the names of people as areas of particular concern. The extent to which the effects of short-term memory deficits on the everyday life of adults with dyslexia are underexplored was highlighted by the authors [32]. More than 25 years later, this remains the case, with little further empirical work on everyday short-term memory having been reported in the interim.

The working memory system is called upon when there is a need not only to store information in temporary memory but also, simultaneously, to manipulate it (e.g., [33,34,35]). This is a particularly important memory system in everyday situations, being involved in, for example, language and reading comprehension (e.g., [36,37]), mental arithmetic (e.g., [38,39]), planning (e.g., [40,41]), and reasoning (e.g., [42,43]). Indeed, it has been argued that working memory deficits are a defining characteristic of dyslexia in adulthood [44]. A meta-analysis of adults with reading disabilities [45] has, likewise, identified problems with short-term and verbal memory as a core area in which difficulties persist into adulthood. Beyond the phonological processing domain, there is evidence from adults with dyslexia that suggests that working memory deficits are modality-general in nature, being most likely to show themselves when task demands are sufficiently taxing [10,11,20,46].

While the effects of dyslexia on working memory have been explored extensively under laboratory conditions, their impact on working memory in everyday life is not well studied and requires a shift to the study of Stanovich’s [22] reflective level of cognition. Some evidence of its effects at the level of typical performance has been gained from self-reports [20] on the self-report Behavior Rating Inventory of Executive Function—Adult Version [47]. On its working memory subscale, the respondents with dyslexia identified more frequent everyday problems in using working memory to maintain information in an active state and to solve problems in a planned and organized way.

Such are the likely effects of dyslexia on volatile, temporary memory in daily life. Long-term memory, on the other hand, is responsible for the encoding, consolidation, and storage of information for later retrieval over longer durations, which may last from a few minutes to decades (e.g., [48,49]). Declarative long-term memory can be subdivided into semantic and episodic memory systems (e.g., [49], but see [50] for a reconsideration of this distinction). Semantic memory consists of memory for facts and transcends time and space [51,52]. In contrast, episodic memory comprises memory for personally experienced events that are located discretely in time and space in an individual’s own past and are autobiographical in nature (e.g., [53,54]).

There is only a small amount of laboratory-based research into long-term memory functioning in adults with dyslexia. However, difficulties in retrieving verbal information from long-term memory have also been documented in adults (e.g., [27]), differences in the way that information is represented in long-term memory [55] and impairments with both item and context memory for verbal information have also been recorded [56]. In children and adolescents, differences in the accuracy or granularity of long-term memories have also been found [6,12,57,58,59].

In everyday life contexts, it has been found [21] that adults with dyslexia self-reported more frequent problems with memory for personally experienced events on the Prospective and Retrospective Memory Questionnaire (PRMQ, [60]). The respondents with dyslexia reported more frequent difficulties over both shorter and longer retention intervals and regardless of whether memory was cued by aspects of the surrounding environment or had to be self-cued. In this study, the proxy-rating PRMQ [61] was also administered to close associates of the PRMQ respondents (such as immediate family or very good friends) to rate the PRMQ respondents using the same questions. The responses of the proxy-rating PRMQ respondents corroborated those of the PRMQ respondents in indicating more frequent retrospective memory problems in the group with dyslexia.

In the context of Higher Education, in-depth interviews with students with dyslexia highlighted the impact of dyslexia-related long-term memory problems when sitting for exams [32]. The students reported that they lacked adequate prompts to help them to recollect pertinent information under examination conditions. They also reported forgetting information under exam pressure that they otherwise knew. Difficulties with memory have also been highlighted in interviews with student teachers [62] and with acting students learning Shakespeare parts [63].

The prospective memory system supports memory for delayed intentions [64] or “remembering to remember” [65]. Prospective memory is called upon whenever there is a delay, no matter how short, between forming an intention to do something and having the opportunity to act upon that intention. Its successful functioning is very important across different settings, ranging from the educational (e.g., remembering to submit coursework by the deadline set), to the workplace (e.g., remembering to send important emails), to the personal or social (e.g., remembering to send a birthday card to a relative or to pay a bill on time). Prospective memory tasks can be habitual (such as remembering to take daily medication at the prescribed times) or episodic, one-off events (such as remembering to meet a friend at a particular café at an agreed time on a specific day) [66]. Prospective memory tasks can also be categorized according to the type of cue that supports prospective remembering (e.g., [67,68]). Event-based cues are provided by objects in the individual’s environment, which serve as reminders of the intention (for example, seeing the shop in question should prompt the individual to enter it to buy a loaf of bread as intended). Action-based cues are also environmentally cued, with the intention being acted upon once a particular activity has been completed. With time-based cues, the individual must remember to carry out an intention at a certain time in the future (for example, returning a telephone call later the same day or paying a credit card bill by the end of the next week). In such cases, the cues to remember the intention need to be self-initiated rather than being prompted by elements existing in the surrounding environment. Instead, the individual must call upon internally generated strategies to ensure that the intention is acted upon at the appropriate time [69].

In adults with dyslexia, lowered accuracy in certain areas of prospective memory has been reported under laboratory conditions [14,15]. While their event-based performance would seem to be at an equivalent level to that of adults without dyslexia, adults with dyslexia have shown significantly worse performance when prospective memory cues are time-based. Evidence of dyslexia-related prospective memory problems has also been found using more naturalistic tasks or requiring tasks to be performed in everyday settings [14,15]. Further to this, aspects of office-based cognition in adults with and without dyslexia have been studied [13], using a non-immersive virtual reality environment to test prospective memory (as well as executive function). Dyslexia-related deficits were found on event- and time-based prospective memory measures, but no impairment was found for action-based prospective memory.

In addition to the long-term memory difficulties documented on the PRMQ [60], more frequent failures of prospective memory in everyday life were also self-reported by the same samples of individuals with dyslexia, both children [16] and adults [21]. Again, the proxy-ratings showed similar group differences in the frequency of prospective memory failures to those obtained in the self-reports. Further to this, it has been found [15] that adults with dyslexia reported experiencing more frequent prospective memory errors on a further self-report measure of prospective memory, the Prospective Memory Questionnaire (PMQ; [70]). The respondents with dyslexia also indicated that they made significantly greater use of the tools and strategies to aid prospective memory on the PMQ’s Techniques Used to Assist Recall scale. After statistically controlling, for this group, the difference in the frequency of use of tools and techniques to support prospective remembering, the adults with dyslexia still highlighted significantly more frequent failures of prospective memory. That failures were still perceived by the adults with dyslexia as being more likely to occur, despite an increased reliance on tools and technology, highlights the importance of understanding their interactions with assistive technology and the goodness of fit between them and the aids that they use to support their everyday cognition and performance. These increased rates of failure occurred when cues to remember had to be generated internally (rather than being evident in the surrounding environment), when the events to be remembered were one-offs, and a longer delay existed between forming the intention and being able to act upon it. When prospective memory demands were short-term and habitual, no differences were found between the two participant groups.

While self-report questionnaires allow some important insights into the daily effects of dyslexia on prospective memory, the voice of the individual adult with dyslexia is still very much lacking, although an interview has been reported with a mother with dyslexia who experienced prospective memory problems when having to remember sequences of childcare tasks that had to be performed after she had returned home from work [71]. She stated that she used lists to help her remember what needed to be bought on the weekly shop but often failed to remember where she had put the list. As a result, she had to rewrite the items that she had already written down, often omitting items that were featured on the original list. In this case, problems with long-term memory compounded her difficulties with an everyday prospective memory task.

Given the evidence of dyslexia-related problems across different memory systems, and the relative lack of direct evidence concerning their everyday impact, qualitative research that prioritizes the participant’s experiences is needed to complement and enrich the extant quantitative data in order better to understand typical levels of performance in daily life. Such qualitative work enables the participant’s voice to be heard. It also allows for the emergence of new information. In the current study, semi-structured, in-depth one-to-one interviews investigated the experiences of adults with dyslexia across different memory systems. Interviewees responded to questions relating to everyday experiences with short-term and working memory, long-term memory, and prospective memory.

In addition to understanding the lived experience of memory in adults with dyslexia, the ways in which memory can be supported with a range of assistive technologies were also explored. Technology was interpreted in a broad sense, ranging from simple tools and objects in the surrounding environment (such as post-it notes) through to information and communication technologies (such as smartphones and tablet computers). The range of mobile technological devices (such as tablet computers and smartphones) available to support adults with dyslexia has been identified [72]. Further to this, a systematic review of inclusive course design for students with dyslexia in Higher Education, which included the use of assistive technologies, has been conducted [73]. Of most relevance to the current study, mixed results were found for the reported effectiveness of recording lectures, with some students finding it useful in reducing the complexity of listening and taking notes at the same time, while others said that they did not have time to listen back to the recordings later. This suggests that using audio recordings as a form of external memory (c.f., [74]) may not be effective. As mentioned previously, adults with dyslexia reported themselves as being more reliant on tools and technology to support their memory than adults without dyslexia [15]). Despite this increased self-reported use, they still identified themselves as experiencing significantly more frequent failures of prospective memory in day-to-day life. This would also seem to suggest that assistive technology is not being used to its full potential (see [75] for a similar point) and also indicates that training in its use should be given so that the benefits to be gained from it are maximized (c.f., [15,76]). However, a resistance to, or disdain for, training reported by university students with dyslexia may present a barrier to the optimal use of assistive technologies [77].

The main aims of the research reported in the current paper were, therefore, twofold. Firstly, the lived experience of different memory systems in adults with dyslexia was explored. Secondly, the research sought to understand the way in which tools and assistive technologies were used by adults with dyslexia to support their memory in both everyday life and in their university studies. Previous research has explored assistive technology in the context of education [72,73,77,78], but the current study added a broader perspective on technology use in adults with dyslexia, probing their experiences across different everyday settings. It also took account of significant technological advances made in the last decade to gain a contemporary perspective on the relationships that adults with dyslexia have with assistive technology.

## 2. Materials and Methods

### 2.1. Participants

Following ethical approval (School of Applied Sciences Ethics Code: SAS1717), the participants were recruited via the Research Participation Scheme at the authors’ host institution, London South Bank University, or in response to advertisements placed around the university. All the interviewees gave informed consent to participate before being interviewed. Twelve female university students with dyslexia (mean age = 32 years and SD = 13.57) took part in the study. Table 1 shows the participants’ pseudonyms, ages, and student status. All the participants were studying Psychology (11 undergraduate and 1 postgraduate) and had English as their first language. Before the interview began, each participant showed the interviewer (EGL) an educational psychologist’s report confirming their dyslexia status or a Disability and Dyslexia Statement from the university prior to being interviewed. The participants received a small honorarium or course credit in appreciation of their time. All names and identifying information have been changed to protect anonymity.

### 2.2. Data Collection

Data were collected using one-to-one semi-structured interviews. Semi-structured interviews are flexible and enable the researcher to follow up or probe interesting areas [79]. Thus, they allow for novel avenues to be explored and a deeper understanding gained of the phenomenon under consideration [80]. An interview schedule was developed as a guide. Open and non-directive questions were used to facilitate discussion, and prompts were included for clarification or to encourage further elaboration. The interview schedule included questions around participants’ views and experiences with different aspects of their memory and about their use and experience of technology as a support for memory in general and their university studies in particular. The interviews were conducted face-to-face on the university campus by EGL and were digitally audio-recorded with the participants’ consent. The interviews lasted between 30 min and 1 h 12 min (mean = 46 min and *SD* = 15.79). At the end of the interview, the participants were given the opportunity to ask any questions and were provided with a debrief form with details of support services. Following the interviews, the recordings were transcribed verbatim.

### 2.3. Data Analysis

The data were analysed using a reflexive thematic analysis [81,82]. This theoretically flexible approach allowed for a detailed analysis of participants’ accounts, unconstrained by prior expectations or assumptions. Initially, the transcripts were read and re-read in order to immerse the analyst researcher (EGL) in the data and become familiar with the participants’ accounts of dyslexia and lived experiences of memory. Coding was then developed by making detailed notes throughout all the transcripts to identify key meanings and points of interest in the data firstly at a semantic, descriptive level and then from a more interpretative stance. The codes were collated in a separate document, and connections and patterns were sought across the codes in order to organize a coherent structure of themes that captured the most salient aspects of the interviews. These themes were reviewed and refined to produce a final table of themes. The analysis was conducted by EGL and discussed with JSS for corroboration or challenge.

## 3. Results

Two overarching themes were identified in the interview data: fallibility of memory and facilitators of memory. A table of themes is presented in Table 2.

### 3.1. Fallibility of Memory

#### 3.1.1. Lack of Trust and Confidence in Memory

The participants expressed a lack of confidence in their memories and concerns that their memories could let them down or lead to embarrassing situations. All participants felt that their memories were poor; for example, Carly stated that her memory is “rubbish”, which is echoed by Kirsten who said, “I’m just rubbish with memory”. Natasha described how “personally I think I’ve got a shocking memory. I wish I had a better memory, I really do”, and Julia related that “I can literally forget anything, sometimes I’ll be making cereal and I’ll take my cereal into the wrong room to get the milk”. The participants also perceived that their memories are poorer in comparison with others. Kirsten shared how she seeks support from others to supplement her memory:

“I think my memory’s very poor, very, very poor. I have to ask my sisters, ‘So what happened then?’ And I go on their memory. I build it up like, ‘and then what happened after that?’, and they say, ‘oh, and what happened then?’ And I go okay.”

Kirsten appeared to have a greater trust in others’ memories and accepted that their recollection of events was more accurate and reliable than her own. A similar account was given by Carly when comparing her memory to that of friends:

“They [friends] remember more detail when they tell a story. My stories are quite brief, theirs are more detailed. Like they can remember the colour of someone’s top like ‘the person in the red top’ and I’m like ‘OK…’ I don’t know, it’s weird, my friends will remember ‘oh the blonde-haired girl’ or something like that and I’m like ‘oh she was blonde?’ (laughter)”

Carly highlighted the greater levels of detail contained in her friends’ memories and again seemed to accept that their memories are more accurate. Julia also reported forgetting more than others and that her memory is not as strong:

“I don’t remember as many things as other people do. So, like me and my sister went travelling together two years ago, and she seems to remember more about it that I do. So, she’ll say, ‘do you remember when we did this or this?’ and I’ll be like ‘no’”.

Here, Julia acknowledged that her sister has a stronger memory of past events they have experienced together of which, at times, Julia has little recollection. Natasha’s lack of confidence in her memory means that she would prefer not to say anything than “embarrass myself by not remembering certain things”.

Leila related how a poor memory permeates into all areas of life: “it is a real disability, forgetfulness, in personal relationships or in jobs, responsibilities”. The forgetting of personal occasions was frequently mentioned by the participants. For example, Sara stated: “I don’t remember birthdays, I don’t remember anyone’s birthday, I don’t even remember my mum’s, my dad’s my sister’s”, a difficulty echoed by Natasha: “oh I forget every birthday (…) I forget all my niece and nephews’ birthdays” and Kirsten: “I don’t really remember the dates of anything really, I have even got my boyfriend’s birthday wrong”. In her social life, Carly recounted difficulties in processing or remembering information, even during relaxed occasions like going to the cinema: “when I watch like a movie or something I won’t know what’s happened, like I always ask my friends ‘what’s happened?’ they’re like ‘I’m watching the same thing as you Carly!’” Following a recipe also presented challenges for several participants, with Natasha sharing “if I have to follow cooking instructions, it’s out the window”. Bea attributed this difficulty to memory deficits:

“It’s actually my short term memory that’s affected, so I’ll be cooking in the kitchen with my boyfriend and he says can you get say 10 g of flour, so I’ll get the flour out but in that really short space of time from him telling me and me actually getting the flour and scales out, I’ve already forgotten what measurement it was meant to be so he has to remind me two or three times before I actually do it”.

Bea also discussed “having no memory for directions” and went on to say, “It’s interesting because I’m a spatial person, but not in that sense. I can’t do that, I’ve been going out with my boyfriend for almost three years, and I still don’t know my way around where he lives, so I can’t use directions and stuff”. Similarly, Kyra struggles with following directions but is able to navigate effectively using landmarks: “If you were telling me, go left, go right, go down, cross the roundabout, I’d find that really hard, but if you said go down to the Asda, turn right and then you’ll come to a school, I’ll remember it because of the places”.

In an academic context, Carly described how “When I don’t remember it’s gone. That’s why I hate exams. When I’m in an exam and I forget it’s very hard to bring it back”. All the participants stated that they did not like exams owing to the pressure of having to recall information and a lack of confidence in their ability to do so. For example, Bea stated, “I don’t like exams because I can’t remember things”, which is echoed by Julia: “I find it really difficult [to recall information], I really don’t like exams, I much prefer coursework”. Natasha went on explain that “I forget basic facts that really should have been drilled into me by now. I know I know it, but I feel like I can’t retrieve the information”.

A lack of confidence in memory can also hinder the participants in expressing themselves; for example, Natasha described the following:

“I struggle with conversations sometimes because I feel like I can’t quickly input into what someone else is saying to me…I’ve got some really great friends and we love having discussions and debates, but I sometimes feel like I can’t get that information out quick enough, so then I feel like I get a little bit, not shy, but maybe don’t vocalise as much as I want to”

Carly also described difficulties with expressing herself:

“If I’m thinking about something and I say it, it’s like it’s different to what I’m thinking in my mind. It just comes out a bit mumble jumbled sometimes (…) I’m like ‘I’m sorry that came out weird’ and other people are like ‘oh it’s not that bad’ but it does come out a bit different”.

Participants expressed a desire to improve their memories and considered that their perceived memory deficits make academic life more difficult and effortful. Pearl stated that “I would just like to remember normal stuff or just the simple things that I actually need to remember, or not have to read thing so many times to have it drilled into my head so that I could then remember”, while Bea said, “I’d really just like to remember things like everyone else does”. There is also a sense in which participants may over-interpret what are the typical memory lapses that most people experience and worry about what constitutes “normal” functioning:

“My short-term memory could be a lot better (…) Especially say I’ve left my room and I’ve forgot something I’d be like ‘oh I just forgot something. Ah damn I’ve got to retrace my steps’ or something like that. But I wish it was better and I wouldn’t be thinking ‘oh have I left something?’ and I wouldn’t be second guessing myself as well.”

Here, Carly described an everyday memory lapse, but it seems to take on additional significance in the context of her perceived memory deficits. She suggested that having a better memory would reduce feelings of self-doubt. This sense of self-doubt also comes through clearly in Bea’s account, whose dyslexia diagnosis amplifies her concerns around her memory: “I don’t actually trust myself a lot of the time, because I’m dyslexic”. Bea’s diagnosis of dyslexia directly diminishes her confidence and trust not only in her memory but also in herself.

#### 3.1.2. Factors Contributing to Memory Failure

Specific factors contributed to the participants’ likelihood of forgetting and to their perceived problems with recall and retention. Participants frequently mentioned that feelings of anxiety, stress, or generally being overwhelmed have a detrimental impact upon their memories and may lead to a spiral of forgetting. For example, Yasmine related how “I kept going around the word to explain to her exactly what I was trying to mean. And then, the more anxious I got, the more I knew it was like it was chasing a balloon, and it was going off”, while for Sara there is “just too much going on and I’m panicking”. Bea also cited stress as a factor in being unable to process information: “If I’m stressed at a train station, say, I cannot read the board to save my life (…) it’s made me so stressed I ended up having a complete mental breakdown because I had no idea where I was meant to be going”. Here, stress contributes to the likelihood of forgetting or difficulties in processing information, which in turn leads to greater stress and panic and further impedance of memory.

Receiving last-minute information or multi-tasking also provided a context for forgetting. When she is able to plan, Kirsten has fewer problems with her memory “but then I always forget if someone throws something in last minute (…) if it wasn’t part of the original plan then it’s gone. You have to message me about seven times before I can remember”. Participants also described instances of forgetting an intention to do something but were reminded through pre-set alarms or calendar entries. For example, Bea shared:

“I have an appointment tomorrow, and I just remembered that today when I looked at my calendar, and then I got an email reminder from them, so it completely slipped my mind, even though I write things on my calendar and have reminders on my phone, because I can’t rely on one thing to remind me. Yeah, I can’t actually remember what I forget.”

Bea’s intention to attend the appointment was only actualised owing to pre-set reminders. To attend her interview, Julia also needed a constant reminder: “like with the room number today, I wrote it on my hand, rather than trying to remember it in my head, as I don’t trust myself to remember it correctly”. When these reminders are not set, participants are more liable to forget appointments or tasks, as recounted by Kyra: “I forgot to unlock the front door for my dad yesterday. I was meant to stay and unlock the front door, but I drove off. I got halfway down the road and my dad was like ‘I’m stuck outside’ I was like ‘Okay, I forgot to unlock the door’”. Kyra’s only prompt to remember here was a phone call from her dad after the event.

Trying to do too many things at once poses problems for Yasmine: “if I go downstairs to the kitchen, I will do other things as well. So, I’ll multitask. When I get there, I’ll do two tasks and then I’ll forget a third one”. Trying to multi-task or deal with competing information also presents difficulties for the participants in an academic context. In particular, listening to a lecture and writing notes simultaneously is considered challenging: “if they’re talking while I write, I’m remembering what I need to write so I’m not taking in what you’re saying” (Pearl). Carly also described a similar difficulty:

“I don’t like to write because by the time I’m writing the first sentence I’ve forgotten what I’m writing and then they’ll be done, and I’ll be like ‘what’s going on?’ I just try and go back and understand what was being said in the lecture rather than going over notes, cos I never understand my notes after (laughter) it’s like half sentences cos I didn’t finish it and I’m trying to keep up”.

Most of the participants said that they did not take notes in lectures owing to slow writing speeds and difficulties in remembering what was being said. For example, Kirsten stated that “I’d still listen to the lecture, but I wouldn’t write anything because my writing is so slow it wouldn’t be worth it”, while Nicole was told in a dyslexia assessment that she has “half the writing speed of an average adult”. Similarly, Meera shared that “I never take notes, never, I just can’t”, which Sara echoed: “I don’t write notes in lectures because I can’t listen and write at the same time, like writing I’m just missing the whole point, I can’t do both”.

The participants also reported needing to repeatedly read materials to make sense of them and to retain information. This was perceived to be time-consuming and effortful. Pearl shared that “to study I need to read certain pages five, six times for it to process in my head and then for me to able to remember bits of it”, while Kirsten feels that exams place her at a particular disadvantage owing to the time it takes to read and process questions:

“In an exam situation even though I get extra time, I can just feel the clock ticking away (…) I mis-read all the time, so I read it like seven times instead of once and it takes so long, you know when you read something so much you don’t actually understand what you’re reading anymore?”

Bea attributed the need to repeatedly read information to her memory rather than specifically to her dyslexia:

“With my dyslexia it’s not that I can’t, so if I look at this, I can read it, the words don’t move, but I cannot actually remember what I read. It’s more about memory than anything else”.

Nicole also associated repeated reading to difficulties with memory and finding the “right words”:

“I get really frustrated with myself that I have to look at something several times because I can’t recall or I just can’t think how to express myself and I can’t think about the real words”.

The participants perceived that their difficulties with memory resulted in them having to work harder than their peers, as articulated by Bea who said, “I just have to work twice as hard…it does take me twice as long to do things because it takes me twice as long to read things, and then it takes me twice as long to actually write out what I’m doing”.

### 3.2. Facilitators of Memory

The participants described various techniques they have developed in order to support their memories. These ranged from pen and paper techniques to use of digital technology.

#### 3.2.1. Preference for Traditional Tools to Support Memory

All the participants stated that they wrote lists and reminders, and used post-it notes, diaries, whiteboards, and calendars as memory tools. Leila is typical of the participants when she said that “I have to write everything down”, which is also echoed by Sara: “I have to write it down, in my notes on my phone, and also on my planning calendar, I just have to write it down somewhere”. The participants also reported writing multiple reminders in different locations; for example, Yasmine stated that “It’s important to write it down. If I constantly write things down over and over again that’s quite a good trigger (…) I’ll put it on the whiteboard at home and I’ve got a big calendar. I will do loads of things”. While the participants used digital technology (e.g., phones and laptops) to write reminders, they all expressed a preference for traditional pen and paper tools. Kirsten described satisfaction in crossing items off her list:

“I just make normal lists on a notepad, I find it easier when I can just tick it off, I feel like I’ve achieved something when I tick it off (…) with a notepad and pen when you cross it off you can still see it whereas on my phone the second you click done, it gets rid of it so it looks like I’ve done nothing”.

Carly also uses lists as reminders for university work:

“I’ll have a whole list of all the things I wanna do this week and then I’ll try and get them all off my list. I’m constantly going back to the list–‘OK, I’ve got this to do, got to do that’ and I’ll put like a Wednesday next to it so this needs to be done by Wednesday. And I always have to look at it every day”.

Yasmine highlighted the physical connection between writing with pen and paper and how this strengthens her memory: “I can picture what I wrote in the lecture, simply because of the colour I use and the way I wrote them down visually, so I’m seeing it visually, I’m using my hand, these are all forms of building up memory for me”. The idea of embodiment strengthening memory is echoed by Julia, who said, “I prefer writing things down (…) I prefer to have it in my planner than on my phone (…) I prefer writing things, cos then I feel like I’m more likely to remember it”. Nicole also linked physically writing out information and seeing it on the page with a greater productivity and ease of processing:

“I prefer paper, I like being able to scribble it out and be able to see that I’ve done an action as well (…) I know that’s so dated, but it just works really. And I think it’s the process of getting the information out of my mind and onto paper, and it’s almost like I’m partly getting there, like getting that action done”.

The participants also spoke about being “visual learners”; for example, Leila states, “I’m a visual and I’m an auditory learner”, and Kyra shared “I am a visual person, so when I revise, I have posters everywhere. So, when I’m sat in an exam something comes up, my mind automatically goes to that poster, so I see the whole thing”. A preference for visual learning links with the techniques frequently used by participants to aid their learning and memory. The use of mind maps and flashcards were mentioned by all the participants. Both Kirsten and Natasha talked about the mental connections that flashcards and mind maps allow them to make: “I did flashcards and that did help because you’ve just got the word associations, just one word triggers all that information (…) it’s helpful because you can visualise it when it comes to remembering” (Kirsten) and “I do like my visual mind map. Only way. I need to see connections to everything”. Colour coding was another technique mentioned by participants as an aid to studying. Kyra said that she puts information “in different colours so I remember it”, and Nicole said, “I’ll colour code them [notes] as well which really helps because then it’s just visual and I can see”.

When preparing for an exam, the participants frequently mentioned rote learning. Meera shared that she repeats information “again and again and again, that really helps a few days before an exam”, and similarly Yasmine said, “I write it down, rote learning, keep writing down over and over and over again, it sticks in and it works”. Carly integrates exam topics into her conversations with friends and family:

“Cos I’m a talker, I tend to talk about it a lot. When I’m in an exam and I forget it’s very hard to bring it back. So, I try to remember conversations I had about it. Rather than, when I read I don’t remember what I read in a sense, so I try to remember what I spoke about to bring it back. So, in exams when I forget what I’m writing I go back to the conversation I had in my front room before [popular TV show] X-Factor or something”.

Talking about exam topics allows Carly to visualize and recall information more easily, possibly because it is rehearsed and retained in a more relaxing environment. Whereas stress and anxiety are detrimental to memory and recall, enjoyment and interest enhance the likelihood of remembering and performing well in assessments, as described by Leila:

“When I have really, really good days, when I don’t have anxiety, and I don’t have stress and I’m not putting too much pressure on myself, it’s a real pleasure to learn, you know it really is, it’s not a struggle. And you realize after you’ve done the assignment, ‘why did I need to make that so difficult for myself?’ Because I was expecting too much of myself”.

Friends and family are also considered important in supporting the participants with their memories. They provide frequent reminders about appointments, dates, and events. This was often carried out because of an awareness of the participants’ problems with memory or previous instances of forgetting. Meera described how her mum is a source of memory support “because she knows I forget things (…) every morning my mum tells me to turn the lights off from outside. I forget, so every morning she has to remind me to do it. If she doesn’t, I won’t do it”. Without reminders from her mum, Meera does not trust that she will remember to do certain tasks. Carly also feels that she will forget things without reminders from others:

“The thing is if they [friends and family] don’t remind me, I’ll forget. I’ll forget, if it’s things like happening during the week, I would forget that then, especially like birthdays and they’ve told me a month in advance and I definitely can’t remember”.

For Leila, reminders from friends and family supplement her use of other tools, which are not wholly reliable when used alone: “People would remind me they’d say ‘Oh, Leila don’t forget about that’ and I’d think, ‘god yeah, thank you for telling me’, even though I’d have it written in my diary I’d forget”.

#### 3.2.2. Use of Digital Technology to Support Memory: Benefits and Limitations

Although the participants generally expressed a preference for traditional pen and paper tools to aid memory, they did also report using digital technology. There did not seem to be any pattern related to age and the likelihood of engaging or not engaging with digital technology. In fact, the younger participants were more likely to describe themselves as “technophobes”. The most commonly used tools were phones and digital recorders. For remembering appointments and social events participants reported using phone reminders, often in combination with entries in diaries and calendars. To remind her of appointments, Yasmine said, “I always have to use the phone. I’m always outside the box, always thinking about something. And I’ve got no sense of, as I always say, no sense of time. So, it’s my biggest crutch”. Nicole also considers her phone a vital memory tool: “I put everything in my phone now, that helps me to remember”. Carly uses her phone in a similar way: “I definitely have to do a reminder on my phone like a little beep the night before or two days before. Yeah, I do little reminders on my phone and write it down”. For Julia, her phone is both a tool to remember appointments and to aid spelling, “if I’m not sure how to spell a word, I say it into my phone or say it into my laptop and it spells it for me”.

The participants had all used digital voice recorders at some point during their university studies. However, they were mixed about how effective they found this tool. For some participants like Leila, digital recorders are “a real essential tool, it’s really important to have available”, while Kyra shared that recorders are “very helpful it just takes the pressure off you trying to remember everything at once, like sometimes it can be a lot of information”. Carly also highlighted that recording technology is “really important because I use it all the time. Because if I don’t use it, I feel like I would get more behind in a sense and not get the grades that I would be getting”. Here, Carly attributed her academic performance to the use of technology and feels she would be at a disadvantage without it. For those participants who found recording useful, having that resource available became indispensable. For example, Nicole describes not attending a tutoring session after forgetting her Dictaphone:

“I wouldn’t go because I didn’t have my Dictaphone (…) because I’ve been there before and I’ve written things down, but I can’t recall it at all and I know I miss some really important information, because they give some really important chunks of information that I can’t record by writing it all down. So, I made my excuses, and I didn’t go”.

Similarly, forgetting her Dictaphone is a source of worry for Leila: “That would make me anxious. I’ve done it before, I’ve gone around lectures saying, ‘oh my god I’ve forgot my Dictaphone would you mind sending it to me?’” The participants who regularly recorded lectures reported almost always listening back to their recordings, usually the same day as the lecture itself. Carly shared that she usually listens back “after the lecture, I do stay after the lecture and seminar’s finished, and I go to the library after and try to go over it. Because if I don’t do it that day it will just be too much”. Leila also shares that she “always” listens back to recordings “so that it stays with me. I benefit more from it and that’s what I’ve been told, with my one-to-one assessment, always make sure you do it more or less the same day, because it does have the tendency to stay more”.

However, not all the participants found recording lectures useful. Julia makes occasional use of recording but also stated that “I don’t always listen back to them. I would only listen back to it if I was revising and if I didn’t understand a certain bit”. Meera also shared a similar experience: “I do sometimes use a voice recorder, but then I wouldn’t really go back to it when I’m going over my lectures”. Kirsten does not consider recording lectures to be helpful, and she doubts the likelihood of re-listening to any recordings made: “if I made my own recording of it, there would be like one bit I need to listen to and I’d need to go through the whole thing and I don’t think I’d do that, I think I’d get bored after trying to find it for ten minutes and I would never listen to it again”.

Following learning needs assessments, the participants had all been offered specialist technology to support them with their learning. The most frequently mentioned were Dragon, ClaroRead, and Sonocent (now Glean). However, while finding some of the basic functions useful, all the participants stated that they did not use this software to its full potential. Both Natasha and Bea described themselves as “technophobes”, while Kyra shared that she is “not technologically savvy”. Navigating the specialist tools was perceived to take a lot of effort and to go beyond what was useful for the participants. For example, Nicole described only using some functions of the specialist software:

“I tend to stick to the basic things, I need to, so there are a number of amazing features that Sonocent might do and I don’t do it, I never look up all of them. And I also have Dragon software which I found useful but the problem with Dragon software is that it’s more sophisticated than it used to be (…) then I become a little jaded with technology and then I stop using it”.

Nicole described becoming overwhelmed by the advanced features of the specialist technology so that she eventually abandons it all together. Carly shared this view of specialist technology as over-complicated and overwhelming:

“I have this laptop and it has this planner, it’s like a spider diagram and you can put it like this happened on this date and what you need to know for it. I find it so complicated sometimes it’s like, it’s quite draining to do it in a sense, it doesn’t really work for me. It is a good idea I think, but I can’t really work it out, so it doesn’t really help me”.

What Carly feels may be a good idea in principle becomes unhelpful owing to how complicated it is, and the effort that would need to be expended to use it is not perceived as worthwhile. Julia also stated that she has “software that I can use, but I find it overcomplicates it. If I’m doing my coursework, I prefer to type it out myself”. Kyra prefers to use the basic functions and straightforward technology: “the recorder is fine, like that’s simple, but the computer ones, I did have someone sit down and show me how to use it because sometimes it can be quite complicated for something really simple, and I would overthink it and I get stressed out”. Overall, participants expressed a preference for easy-to-use technology.

## 4. Discussion

The current study was undertaken to gain insights into the everyday experience of memory in female university students with dyslexia. As noted in the Introduction, memory in adults with dyslexia has been researched to varying degrees under laboratory conditions (e.g., [10,11,14,15]), with naturalistic (e.g., [14,15]) and self-report questionnaire work (e.g., [15,19,21], providing some understanding of the ways in which dyslexia-related memory problems can play out in real-world settings. To give more depth to this understanding, a qualitative, semi-structured interview approach was used in the current study to explore memory functioning. The participants’ experiences were probed across different systems (short-term/working memory, long-term memory, and prospective memory) and different settings (home, social, educational, and work). From the thematic analyses carried out on the 12 interviews, two overarching themes were identified. The first of these, fallibility of memory, was further subdivided into two sub-themes, relating to (i) a lack of trust and confidence in memory and (ii) the factors contributing to memory failure. The second theme, facilitators of memory, was also broken down into two sub-themes, (i) a preference for traditional tools to support memory and (ii) the use of digital tools to support memory: benefits and limitations. These themes and sub-themes will be considered in more depth and related to the literature on memory in dyslexia in the following paragraphs.

While it should be acknowledged that all adults experience everyday memory failures to differing frequencies and degrees, the participants clearly felt that their memory abilities were inferior to those of family members and friends who did not have dyslexia. They reported trusting the memories of other people over their own. Difficulties were described across different memory systems, and interactions with both executive function and emotion can be identified in their interviews. These problems were experienced across different settings. Moreover, meta-awareness of the fallibility of their memory led to a lack of confidence in their memory being voiced by participants. This lack of confidence, in turn, seems to have then made their cognition more vulnerable to the deleterious effects of stress and other emotions. The impact of this increased vulnerability in educational settings was highlighted by the participants, particularly in examination settings where optimal memory performance is required. The participants also identified the consequences of this lack of confidence in social settings, citing difficulties in expressing themselves and a lack of confidence in their memory impeding their ability to contribute to discussions with their friends about social events. Set against these negative consequences, however, the participants also identified the benefits to remembering bestowed by their enjoyment and motivation. The participants showed an awareness of the shortcomings in their memory functioning and indicated the ways in which they compensated for these difficulties.

Regarding short-term and working memory, difficulties were raised by the participants with following recipes, updating memory in the light of new information becoming available (shown both when cooking and in following films), and following directions to places. The participants also identified problems with multi-tasking both when attending lectures (finding it difficult or impossible to take notes and listen at the same time) and at home (when trying to complete several tasks at the same time). They also highlighted difficulties in keeping track of plans (such as when following recipes). These problems link not only to the dyslexia-related working memory difficulties documented under laboratory conditions (e.g., [11,20]) but are also consistent with theoretical accounts of dyslexia that argue for attentional allocation difficulties [83] and problems with goal maintenance (e.g., [11,84]).

In the case of long-term memory, the participants reported experiencing greater difficulties with both declarative and episodic memory in comparison with their family and peers. In educational settings, such as university, the participants voiced the frustration that difficulties with declarative long-term memory meant that they had to work considerably harder than their peers without dyslexia to achieve the same ends (see [85] for similar perceptions arising from interviews with adults with dyslexia). This feeling was expressed particularly in relation to examinations. A concern about remembering material under examination conditions also arose from previous interviews with university students with dyslexia [32]. The need for repetition of information to commit it to memory had also come out of interviews with registered nurses with dyslexia [86], but again, with levels of concern over the fallibility of their memories being such that they often wrote information down to act as prompts. In lectures, some participants reported supporting their long-term memory through using voice recorders (in addition to written notes). However, other participants deemed such recordings to be of limited value, highlighting the cost/benefit trade-off in searching through audio files to locate the exact information that they needed. This is also consistent with the previous literature [73]. In social contexts, the participants identified the difficulties associated with an increased difficulty with remembering birthdays and other important dates (and despite social motivation to remember being high). Problems with the encoding of information were identified by the participants when they were feeling stressed, under time pressure, or were trying to multi-task (c.f., [83,87]).

A lack of detail surrounding long-term episodic memories of events shared with family or friends was highlighted by several participants. They reported a reduced level of granularity to the information that they could recall in comparison with the more detailed recollections of those around them. Problems with long-term memory in adults with dyslexia have also been identified in questionnaire work [21], while laboratory-based research has also identified differences in long-term memory representations of adults with and without dyslexia [55]. Whether this reduced level of detail occurs at the time of encoding or whether it is a failure of maintenance or access is an open question and could be a fruitful avenue of future research. Certainly, in terms of declarative memory, it appears that the participants required multiple readings of material for it to be encoded effectively. Particular difficulties with the efficient and flexible access of declarative information that had successfully been stored in long-term memory came out in several interviews. These problems in accessing information are very suggestive of the difficulties with verbal fluency that have been documented under experimental conditions in adults with dyslexia (e.g., [30,88,89,90,91]). The participants in the current study reported slower, more effortful, and reduced access to information in long-term memory, all hallmarks of reduced verbal fluency (e.g., [92,93,94]). Difficulties were also reported by some participants in organizing the information that they accessed in order to express it clearly to others. Dyslexia-related problems with planning and organization have been identified in children [95,96,97] and adults with dyslexia (e.g., [13,98,99]).

In relation to prospective memory, the participants identified problems particularly in home and social settings, such as remembering to attend appointments or turn off lights as they had intended. Again, a lack of trust in their own ability to remember to carry out actions as intended was identified by several participants. Some of the examples reported by the participants were similar to the difficulties highlighted previous self-report research [19] using the Cognitive Failures Questionnaire [100], particularly around absentmindedness.

The participants also highlighted how they frequently forgot their intentions but were saved by technological prompts. By their very nature, prospective memory tasks particularly lend themselves to support by tools and technology (e.g., see a systematic review and meta-analysis [101]). Indeed, the participants reported using a range of such supports for their prospective memory, and this was the case even for routine prospective memory tasks rather than more cognitively challenging one-off tasks. They identified the use of lists, post-it notes, whiteboards, and calendars to support their prospective remembering and also identified the effective visual support that the use of colour coding and mind maps conferred (see [75] for similar reports of the benefits of visual study aids). In addition to the use of audio recordings to support long-term memory that has already been considered in this section, the participants identified other ways in which they used tools and technology to support their everyday memory function. While providing considerable benefits to some participants, costs were also identified in terms of an over-reliance on technology and a feeling of powerlessness in its absence if forgotten. The participants also identified issues with the complexity of some technology, preferring low-tech solutions (such as pen and paper) and raising issues over ongoing support for software applications if their use is to be continued over time. Similar issues over training and support for adults with dyslexia have been identified previously (e.g., [15,76]). Adults with dyslexia and individuals supporting them need to be aware that proactivity is needed on both sides to ensure that technology is being used effectively and that its use is maintained over time.

Aspects of embodied cognition (e.g., [102,103], see [104], for a discussion of embodied cognition in STEM learning) and distributed cognition (e.g., [74,105]) were raised by the participants. The participants reported the strengthening of memory traces through embodied action, such as writing notes and lists by hand, rather than entering them into electronic devices. In many cases, family and friends of students with dyslexia seemed to provide an invaluable source of memory support. Their social networks seem to act as a form of distributed cognition (c.f., [74,106]), reminding the participants of upcoming social events and allowing them to offload aspects of the planning and organizing onto other people. Again, there are resonances with previous qualitative research involving adults with dyslexia. In healthcare work, informal cognitive support provided by co-workers has been identified [107,108]. In Higher Education, the supportive role that social networks play for university students with dyslexia has been highlighted [109]. Finally, in a home context, they are also consistent with previous work in which an adult with dyslexia reported relying on a partner for time management and getting him to appointments in time [71]. Given that these social sources of support were used in conjunction with tools and technology, the current study gives interesting insights into the rich and complex distributed cognitive systems of adults with dyslexia (c.f., [105]).

One limitation of the current study is that it relies on self-reports of subjective experiences and perceptions which, moreover, may be coloured by the problems with self-esteem, which often accompany the individual with dyslexia into adulthood (e.g., [32,110,111,112]). However, previous self-report questionnaire research has indicated that, compared with adults without dyslexia, adults with dyslexia do not hold an unduly negative or inconsistent view of their cognitive abilities [20]. This makes it more likely that if difficulties are self-identified by adults with dyslexia, then they do, in reality, exist. Further, questionnaire work has also indicated that cognitive difficulties are more frequently identified in adults with dyslexia by people who are in close contact with them rather than by the individuals themselves [19,21]. The sample consisted entirely of female participants, and this can be considered both a strength (in terms of a greater homogeneity of experience) and a limitation (in that the voices of adult males with dyslexia are not being heard). However, it should be noted that, historically, dyslexia has been considered a neurodevelopmental condition that affects males more frequently than females partly due to a referral bias and partly due to a female advantage in language skills from an early age [113]. There is, thus, much to be gained from a greater insight into how dyslexia affects women in day-to-day life. Further research could be conducted to explore the lived experience of men with dyslexia to address this limitation (such as it is). A further limitation is that, as this was a qualitative study of lived experience, no comparison group was employed. Set against this potential criticism, the approach adopted in the current study did not seek to make comparisons but to focus on the experiences of a fairly homogenous group of participants. The importance of giving an individual voice to adults with dyslexia should not be overlooked, and moreover, the findings arising from the current study allow triangulation with the results of studies using experimental or self-report methodologies. As has been argued in this section, the parallels between the lived experiences of the participants and both experimental and self-report questionnaire research are very much evident and lend support to their validity (e.g., [11,14,15,19,20,21]). However, a future study could be designed to develop upon the approach taken in the current paper to involve a comparison group, if it fitted with the researcher’s philosophical approach to qualitative methods. While the older interviewees in the current sample did not voice any particular negativity around the adoption and use of digital technology (and, indeed, younger interviewees were likely to identify themselves as technophobes), it would be beneficial to consider the needs of the growing demographic of older adults with formal diagnoses of dyslexia in future research and what influence the interaction of dyslexia and age might have on the adoption and use of new digital assistive technology. Just as the effects of dyslexia do not end with the onset of adulthood (e.g., [44]), the need for continued support into later working age and into retirement should be explored in future research, especially since the demands on cognition may differ from those of earlier adulthood. Finally, the issues of both defining and diagnosing dyslexia are complex (e.g., [114]), with the severity of core symptomatology and possible cognitive subtyping (e.g., [115,116,117]) to consider. It might prove fruitful for further research to explore these issues and their possible influence on the type and frequency of everyday memory failure in adults with dyslexia, although such investigations might better suit a quantitative methodology.

The current qualitative study of the lived experiences of students with dyslexia highlights the many and varied challenges that they face in everyday situations across different settings and varying memory domains. While previous qualitative research has explored the experience of adults with dyslexia, this has usually been in the context of either education (e.g., [32,75,107,118,119]) or work (e.g., [107], for a systematic review of qualitative studies, see [120]) and has also tended to focus on socio-emotional aspects. To the best knowledge of the authors, the current study is novel in focusing specifically on the everyday memory functioning of adults with dyslexia from a qualitative perspective. While differing in focus, the reports of the impact of dyslexia on memory are consistent with those identified in previous qualitative research (e.g., [32,62]) and those documented in self-report questionnaire studies (e.g., [15,21]) and (semi-)naturalistic research (e.g., [14,15]. These different sources of evidence should triangulate to inform support arrangements and reasonable adjustments. Avenues for future cognitive psychological research have been identified through talking to students with dyslexia about their experiences as well as useful insights into their use of technology, which could be considered by software application developers.

Given the memory concerns highlighted by the interviewees and their likely negative effects on everyday cognition, the applicability and potential utility of memory training programs, such as Lindamood-Bell [121], the Turnabout Programme [122], or Pearson’s Cogmed Working Memory Training [123] could be considered by adults with dyslexia and the individuals supporting them in educational and employment settings. While such programs might have benefits for adults as well as children with dyslexia, it should be noted that the effectiveness, long-term benefits, and near- and far-transfer of memory training approaches are continued sources of discussion in the literature (e.g., [124,125,126,127]). Further to these formalized memory training programs, metacognitive strategies by which prospective memory might be improved in dyslexia have been identified and considered in previous work [76].

In conclusion, the qualitative data reported in the current paper lend support, and add greater depth, to the quantitative results obtained previously from laboratory-based research, naturalistic studies, and self-report questionnaires (e.g., [10,11,13,14,15,19,20,21,46,56,57,58,59]). They indicate the ways in which memory difficulties permeate the daily lives of adults with dyslexia across different settings, giving important insights into the lived experience of adults with dyslexia when using their memory systems in everyday settings and in response to day-to-day tasks. It is clear from the interviews that the challenges faced by adults with dyslexia extend beyond the reading and spelling difficulties that are typically associated with the condition. Theoretical accounts of dyslexia need to explain broader cognitive difficulties within their frameworks. To provide better targeted support for adults with dyslexia, a greater awareness of the potential impact of these difficulties in educational, workplace, and social settings is required.

## Figures and Tables

**Table 1 behavsci-13-00840-t001:** Participant details.

Participant Pseudonym	Age	Student Type ^1^
Nicole	42	PG
Kyra	27	UG
Carly	24	UG
Leila	63	UG
Bea	22	UG
Yasmine	55	UG
Meera	21	UG
Kirsten	20	UG
Pearl	27	UG
Natasha	34	UG
Julia	21	UG
Sara	32	UG

^1^ Key: UG = undergraduate; PG = postgraduate.

**Table 2 behavsci-13-00840-t002:** Themes and sub-themes.

Theme	Sub-Theme
Fallibility of memory	Lack of trust and confidence in memory Factors contributing to memory failure
Facilitators of memory	Preference for traditional tools to support memory Use of digital tools to support memory: benefits and limitations

## Data Availability

The data presented in this study are available on request from the corresponding author. The data are not publicly available due to their containing information that might compromise the anonymity and privacy of the interviewees.

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
