# Peer review of "Lived Experiences of Everyday Memory in Adults with Dyslexia: A Thematic Analysis"

_behavsci, 2023, doi:10.3390/bs13100840_

Round 1
Reviewer 1 Report
The authors present a qualitative research study based on semi-structured interviews investigating lived experiences of everyday memory in adults with dyslexia. It is argued that there is considerable value to be gained from documenting the lived experiences of adults with dyslexia, given the potential negative implications of memory problems across educational, work, social, and personal settings. In doing so, the authors argue, it should be possible to identify the areas in which there will be a greater likelihood of error by adults with dyslexia and for these areas to be officially recognized in support arrangements.
The study is well-presented, well-written and the authors’ line of reasoning is easy to follow. The authors’ interpretation of the transcribed interview material appears balanced and unbiased. Overall, it is a well-executed piece of qualitative research which gives voice to 12 adult females with dyslexia and their challenges and struggles with memory-related difficulties.
Here are some brief comments based on my reflections upon reading the manuscript.
1. It would benefit the manuscript if the notion of dyslexia was discussed in some more detail. It is a complex diagnosis and there is no single generally accepted definition. Evidence suggests that dyslexia is not an all-or-none phenomenon but rather occurs with varying degrees of severity. Here, since the matter is not discussed, readers might easily get the wrong perception. Unfortunately, we receive no information about how the study participants were diagnosed, what type of tests were involved in their assessment, and which criteria were applied to give the diagnosis. Also, many researchers argue that there are various subtypes of dyslexia. Was such information available about the study participants? If so, it might be interesting to try and understand if different subtypes or nuances of dyslexia could be differentially related to the memory difficulties experienced.
2. The age span of the participants is rather wide (20-63). This makes me wonder whether there were any discernable trends relating to age in the material. This is not discussed at all I believe. For example, the age of the study participants might potentially influence their inclination to use digital technology to support memory, above and beyond their diagnosis of dyslexia. So, discussing the participants’ use of modern technology without taking age into account may potentially be misleading.
3. Even for a person without dyslexia, many of the sentiments expressed by the participants when discussing fallibility of memory are relatable and recognizable from one’s own experience. It is therefore somewhat difficult to understand what differentiates the memory experiences of these individuals from individuals without dyslexia. Personally, I believe that this speaks to the need for a comparison / control group in future studies of a similar kind.
Author Response
We thank the reviewer for their helpful and interesting comments. We have reproduced each comment below and provided an answer to it directly afterwards.
It would benefit the manuscript if the notion of dyslexia was discussed in some more detail. It is a complex diagnosis and there is no single generally accepted definition. Evidence suggests that dyslexia is not an all-or-none phenomenon but rather occurs with varying degrees of severity. Here, since the matter is not discussed, readers might easily get the wrong perception. Unfortunately, we receive no information about how the study participants were diagnosed, what type of tests were involved in their assessment, and which criteria were applied to give the diagnosis. Also, many researchers argue that there are various subtypes of dyslexia. Was such information available about the study participants? If so, it might be interesting to try and understand if different subtypes or nuances of dyslexia could be differentially related to the memory difficulties experienced.
- As stated in the Participants subsection (Lines 249-252 of the original manuscript), the interviewees showed the interviewer (EL) a report from an educational psychologist confirming their diagnosis. The tests making up these diagnoses vary between individuals, meaning that they are not open to comparison. Further to this, the educational psychologists’ reports indicated an overall diagnosis of dyslexia but without specifying particular subtypes. The focus of the research was on the lived experience of students who identified as having dyslexia (with this being backed up by a formal diagnosis by a suitably qualified professional). However, we agree that examining the issues of both dyslexia severity and subtyping and their relative impact on everyday memory would be interesting to explore. We have thus incorporated these ideas into the limitations and suggestions for further study presented in the Discussion, with a brief acknowledgement of these at the start of the Introduction. We thank the reviewer for making this point.
- The age span of the participants is rather wide (20-63). This makes me wonder whether there were any discernable trends relating to age in the material. This is not discussed at all I believe. For example, the age of the study participants might potentially influence their inclination to use digital technology to support memory, above and beyond their diagnosis of dyslexia. So, discussing the participants’ use of modern technology without taking age into account may potentially be misleading.
- Three older Ps are identified in the Participants table (Leila, Nicole, and Yasmine). We have cross-checked their names against the comments on for digital technology. It did not emerge as a subtheme from the analysis and, if anything, it was the younger interviewees who identified as technophobes. We have now added information on this in the Analysis section to contextualise the comments and indicate that they were not age-related. The general psychological literature on ageing would typically consider “old” adults to be aged 65 years and over. A search of the literature on technological adoption would seem to consider 45 years as the youngest age at which differences in age might appear. This is an interesting point to consider in supporting older adults with dyslexia, as more such individuals with formally recognised diagnoses reach later working age and retirement age. We have thus noted it in the Discussion.
- Even for a person without dyslexia, many of the sentiments expressed by the participants when discussing fallibility of memory are relatable and recognizable from one’s own experience. It is therefore somewhat difficult to understand what differentiates the memory experiences of these individuals from individuals without dyslexia. Personally, I believe that this speaks to the need for a comparison / control group in future studies of a similar kind.
- We agree with the reviewer and have acknowledged that these types of error occur to all people in Lines 649-651 of the original manuscript. It is also addressed in the original manuscript (Lines 783-787), as well as being alluded to earlier in the same paragraph when comparing the results with self-report studies where a comparison group (matched for age and short-form IQ) was used (including Lines 766-769). As we note, the use of a control group would not sit comfortably with our interpretation of the philosophy of qualitative research. However, we have incorporated the reviewer’s concern about a comparison group being interviewed in future qualitative research.
Reviewer 2 Report
Dear Authors, thank you for your paper. I've read it very careful. I think that the paper is very useful to support adults with dyslexia.
Please, in line 518, check spelling (repat --> repeat?).
It would be very useful if you add some information about the rehabilitation of memory issues for adults with dyslexia (i.e. Lindamood Bell, Turnabout program..).
Author Response
We thank the reviewer for their helpful and interesting comments. We have reproduced each comment below and provided an answer to it directly afterwards.
Please, in line 518, check spelling (repat --> repeat?).
- Thank you for noticing this typographical error. It has been corrected to “repeat”.
It would be very useful if you add some information about the rehabilitation of memory issues for adults with dyslexia (i.e. Lindamood Bell, Turnabout program..).
- We thank the reviewer for this suggestion and have added a paragraph towards the end of the Discussion to briefly discuss this issue.
Reviewer 3 Report
Thank you for allowing me to review this manuscript. The background knowledge supports this topic very well and the current qualitative research reported here not only supports the quantitative works, it personalizes the lived experiences reminding the readers of the actual affect.
Very important is the message that comes from the reading but is not explicitly stated. With effort and exploration of a variety of tools, individuals with dyslexia can succeed equally to their peers and should not be subjected to lower expectations during the primary and secondary educations.
I find two statements in this paper that do not belong because they are not the questions asked, nor does the research support them. These statements are 'solution' statements or perhaps opinion statements.
The first statement to be reconsidered and perhaps argued in a future paper: Page 14 - Lines 684-689:
This feeling was expressed particularly in relation to examinations and highlights the need for inclusive assessment strategies to ensure fairness. Based on the current interviews, being allowed more time in an exam itself (as is standard in UK universities) would seem to be no adequate compensation for the likely extra hours put into preparing for the exam, reading over material again and again until it “sticks”.
This would be a stronger and applicable statement if the authors used this to explained that hard work pays off. That by using these tools, these adults are successful. It is not a bad thing to struggle in life to attain knowledge or success. Professional athletes put in way more time and effort than others to gain success. The idea that we should accommodate so students do not have to work so hard to learn and gain an equal degree should not be the message.
However, the main point here is that one statement of accommodations does not fit into the very well written manuscript.
The second statement to be reconsidered to be reworded:
Page 15 - Lines 744-748
The participants also identified issues with the complexity of some technology, preferring low-tech solutions (such as pen and paper) and raising issues over ongoing support for software applications if their use is to be continued over time. Similar issues over training and support for adults with dyslexia have been identified previously.
This statement may just need to be rewritten in such a way that suggests individuals with dyslexia may need to seek out training and support for technology difficulties and updates. Written as it is here, the authors allude to the idea that an outside source, someone other than the adult with dyslexia should provide the support.
NOTE: In our world of high-tech, low-tech solutions are still extremely valid...sometimes the internet or power goes out or devices are left behind, thus hand-written notes put in highly visible locations should be encouraged, not discouraged.
I have not only enjoyed reading this manuscript but learned a good bit! I teach this topic to future teachers, and I have many students come through with dyslexia and much appreciate this information!! Thank you for allowing me to be part of this review process.
Excellent paper!
Author Response
We thank the reviewer for their helpful and interesting comments and are very pleased to hear how much they enjoyed reading the paper. We have reproduced each comment below and provided an answer to it directly afterwards.
The first statement to be reconsidered and perhaps argued in a future paper: Page 14 - Lines 684-689:
This feeling was expressed particularly in relation to examinations and highlights the need for inclusive assessment strategies to ensure fairness. Based on the current interviews, being allowed more time in an exam itself (as is standard in UK universities) would seem to be no adequate compensation for the likely extra hours put into preparing for the exam, reading over material again and again until it “sticks”.
This would be a stronger and applicable statement if the authors used this to explained that hard work pays off. That by using these tools, these adults are successful. It is not a bad thing to struggle in life to attain knowledge or success. Professional athletes put in way more time and effort than others to gain success. The idea that we should accommodate so students do not have to work so hard to learn and gain an equal degree should not be the message.
- While such perseverance and dedication is clearly laudable, the point that we were making was about equity and fairness and whether it is right as educationalists to expect adults with dyslexia to put in more exam revision hours than their peers without dyslexia when there are alternative methods of assessment that do not exacerbate their difficulties (with revision, for example, we are likely to see problems not only from slower and less accurate reading of revision material but also difficulties then with encoding it in memory). The perception of the interviewees was that they were having to work harder than their peers for the same results. The transcripts themselves do not provide evidence either way as to whether their efforts lead to greater or lesser levels of success. The most that the data say is that they “achieve the same ends” through more time spent revising. We have removed the sentence in question, thereby sticking more closely to the perceptions of the interviewees and avoiding straying too far into supposition.
However, the main point here is that one statement of accommodations does not fit into the very well written manuscript.
The second statement to be reconsidered to be reworded:
Page 15 - Lines 744-748
The participants also identified issues with the complexity of some technology, preferring low-tech solutions (such as pen and paper) and raising issues over ongoing support for software applications if their use is to be continued over time. Similar issues over training and support for adults with dyslexia have been identified previously.
This statement may just need to be rewritten in such a way that suggests individuals with dyslexia may need to seek out training and support for technology difficulties and updates. Written as it is here, the authors allude to the idea that an outside source, someone other than the adult with dyslexia should provide the support.
- We agree that the issue of personal agency is not expressed unambiguously here. We have adjusted the text to acknowledge roles both for students with dyslexia and their study advisors (or workplace advisors) in ensuring that training and support are maintained in line with technological developments or difficulties. As adults, people with dyslexia should be reaching out for support but they may not do so due to self-esteem problems (or just a simple lack of time). We do not feel that advisors recommending assistive technology solutions should absolve themselves of responsibility once the recommendation has been made. Proactivity on both sides would be ideal in identifying and fixing problems – this would have obvious benefits for both the individual with dyslexia and the taxpayer (if equipment is bought through local or central government support schemes). We have added to the text to indicate that agency is required by both adults with dyslexia and people supporting them.
NOTE: In our world of high-tech, low-tech solutions are still extremely valid...sometimes the internet or power goes out or devices are left behind, thus hand-written notes put in highly visible locations should be encouraged, not discouraged.
- We thank the reviewer for this comment. We absolutely agree, use such methods ourselves, and are not aware of having discouraged the use of low-tech solutions in the manuscript. The interviewees themselves professed a preference for such solutions and this is reported as a sub-theme for Theme 2 (facilitators of memory). We have re-read the manuscript and do not think that we have been judgemental about such methods or their relative merit compared with high-tech solutions. We are firmly of the view that whatever the individual with dyslexia feels works for them should be what they use.